# GENERATING DIVERSE HIGH RESOLUTION IMAGES WITH VQ-VAE

**Ali Razavi**[*]
DeepMind
alirazavi@google.com

**Aäron van den Oord**[*]
DeepMind
avdnoord@google.com

**Oriol Vinyals**
DeepMind
vinyals@google.com

## ABSTRACT

We explore the use of Vector Quantized Variational AutoEncoder (VQ-VAE) models for large scale image generation. To this end, we scale and enhance the autoregressive priors used in VQ-VAE to generate synthetic samples of much higher coherence and fidelity than possible before. We use simple feed-forward encoder and decoder networks, thus our model is an attractive candidate for applications where the encoding and decoding speed is critical. Additionally, this allows us to only sample autoregressively in the compressed latent space, which is an order of magnitude faster than sampling in the pixel space, especially for large images. We demonstrate that a multi-scale hierarchical organization of VQ-VAE, augmented with powerful priors over the latent codes, is able to generate samples with quality that rivals that of state of the art Generative Adversarial Networks on multifaceted datasets such as ImageNet, while not suffering from GAN's known shortcomings such as mode collapse and lack of diversity.

## 1 INTRODUCTION

Deep generative models have significantly improved in the past few years [1; 18; 17]. This is, in part, thanks to architectural innovations as well as computation advances that allows training them at larger scale in both amount of data and model size. The samples generated from these models are hard to distinguish from real data without close inspection, and their applications range from super resolution [15] to domain editing [32], artistic manipulation [24], or text-to-speech and music generation [17].

We distinguish two main types of generative models: maximum likelihood based models, which include VAEs [11; 21], flow based [5; 20; 6; 12] and autoregressive models [14; 27]; and implicit generative models such as Generative Adversarial Networks (GANs) [8]. Each of these models offer several trade-offs such as sample quality, diversity, speed, etc.

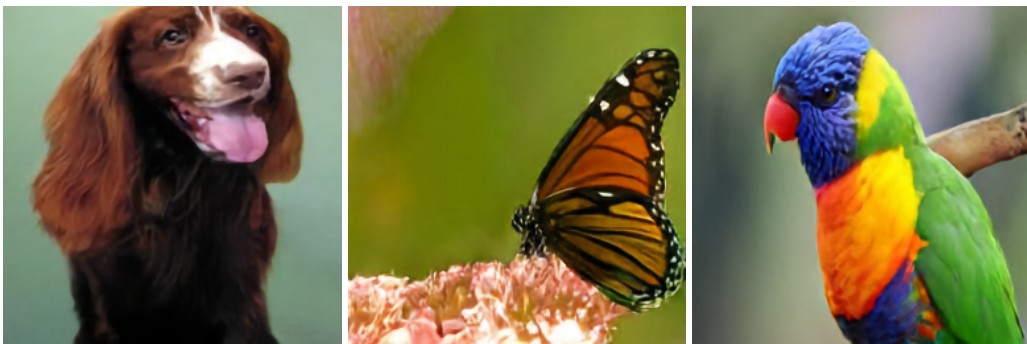

Figure 1: Class-conditional 256x256 image samples from a two-level model trained on ImageNet.

---

[*]Equal contributions.

GANs optimize a minimax objective with a generator neural network producing images by mapping random noise onto an image, and a discriminator defining the generators' loss function by classifying its samples as real or fake. Larger scale GAN models can now generate high-quality and high-resolution images [1; 10]. However, it is well known that samples from these models do not fully capture the diversity of the true distribution. Furthermore, GANs are challenging to evaluate, and a satisfactory generalization measure on a test set to assess overfitting does not yet exist. For model comparison and selection, researchers have used image samples or proxy measures of image quality such as Inception Score (IS) [23] and Fréchet Inception Distance (FID) [9].

In contrast, likelihood based methods optimize negative log-likelihood (NLL) of the training data. This objective allows model-comparison and measuring generalization to unseen data. Additionally, since the probability that the model assigns to *all* examples in the training set is maximized, likelihood based models, in principle, cover all modes of the data, and do not suffer from the problems of mode collapse and lack of diversity as seen in GANs. In spite of these advantages, directly maximizing likelihood in the pixel space can be challenging. First, NLL in pixel space is not always a good measure of sample quality [25], and cannot reliably be used to make comparison between different model classes. There is no intrinsic incentive for these models to focus on, for example, global structure. Some of these issues are alleviated by introducing inductive biases such as multi-scale [26; 27; 19; 16] or by modeling the dominant bit planes in an image [13; 12].

In this paper we use ideas from lossy compression to relieve the generative model from modeling negligible information. Indeed, techniques such as JPEG [31] have shown that it is often possible to remove more than 80% of the data without noticeably changing the perceived image quality.

As proposed by [29], we compress images into a discrete latent space by vector-quantizing intermediate representations of an autoencoder. These representations are over 30x smaller than the original image, but still allow the decoder to reconstruct the images with little distortion. The prior over these discrete representations can be modeled with a state of the art PixelCNN [27; 28] with self-attention [30], called PixelSnail [3]. When sampling from this prior, the decoded images also exhibit the same high quality and coherence of the reconstructions (see Fig. 1). Furthermore, the training and sampling of this generative model over the discrete latent space is also 30x faster than when directly applied to the pixels, allowing us to train on much higher resolution images. Finally, the encoder and decoder used in this work retains the simplicity and speed of the original VQ-VAE, which means that the proposed method is an attractive solution for situations in which fast, low-overhead encoding and decoding of large images are required.

## 2 BACKGROUND

### 2.1 VECTOR-QUANTIZED AUTO-ENCODERS

The VQ-VAE model [29] can be thought of as a communication system. It consists of an encoder that maps observations onto a sequence of discrete latent variables, and a decoder that reconstructs the observations from the discrete code. Both use a shared codebook. The encoder is a non-linear mapping from the input space, $x$, to a vector in an embedding space, $E(x)$. The resulting vector is then quantized based on its distance to the prototype vectors in the codebook $c_k, k \in 1 \ldots K$ such that each vector $E(x)$ is replaced by the index of the nearest prototype vector in the codebook and is transmitted to the decoder (note that this process can be lossy). The decoder maps back the received indices to their corresponding vectors in the codebook, from which it reconstructs the data via a non-linear function. To learn these mappings, the gradient of the reconstruction error is then back-propagated to the decoder, and to the encoder using the straight-through gradient estimator. The VQ-VAE model incorporates two additional terms in its objective to align the vector space of the codebook with the output of the encoder.

The *codebook loss*, which only applies to the codebook variables, brings the selected codebook $c$ close to the output of the encoder, $E(x)$. The *commitment loss*, which only applies to the encoder weights, encourages the output of the encoder to stay close to the chosen codebook vector to prevent it from fluctuating too frequently from one code vector to another. The overall objective is described in equation 1, where $c$ is the quantized code for the training example $x$, $E$ is the encoder function and $D$ is the decoder function. The operator $sg$ refers to a stop-gradient operation that blocks gradients from flowing into its argument, and $\beta$ is a hyperparameter which controls the reluctance to change

the code corresponding to the encoder output.

$$L = ||x - D(c)||_2^2 + ||sg[E(x)] - c||_2^2 + \beta ||sg[c] - E(x)||_2^2 \qquad (1)$$

As proposed in [29], for the codebook loss (the second term in equation 1) we use the exponential moving average updates for the codebook, as a replacement for the second loss term in Equation 1:

$$N_i^{(t)} := N_i^{(t-1)} * \gamma + n_i^{(t)}(1 - \gamma) \qquad (2)$$

$$m_i^{(t)} := m_i^{(t-1)} * \gamma + \sum_j E(x)_{i,j}^{(t)}(1 - \gamma) \qquad (3)$$

$$e_i^{(t)} := \frac{m_i^{(t)}}{N_i^{(t)}}, \qquad (4)$$

where $n_i^{(t)}$ is the number of vectors in $E(x)$ in the mini-batch that will be quantized to codebook item $e_i$ and $\gamma$ is a decay parameter with a value between 0 and 1. We used the default $\gamma = 0.99$ in all our experiments. We use the released VQ-VAE implementation in the Sonnet library [1] [2].

## 3 METHOD

The proposed method follows a two-stage approach: first, we train a hierarchical VQ-VAE (see figure 2) to encode images onto a discrete latent space, and then we fit a powerful PixelCNN prior over the discrete latent space induced by all the data.

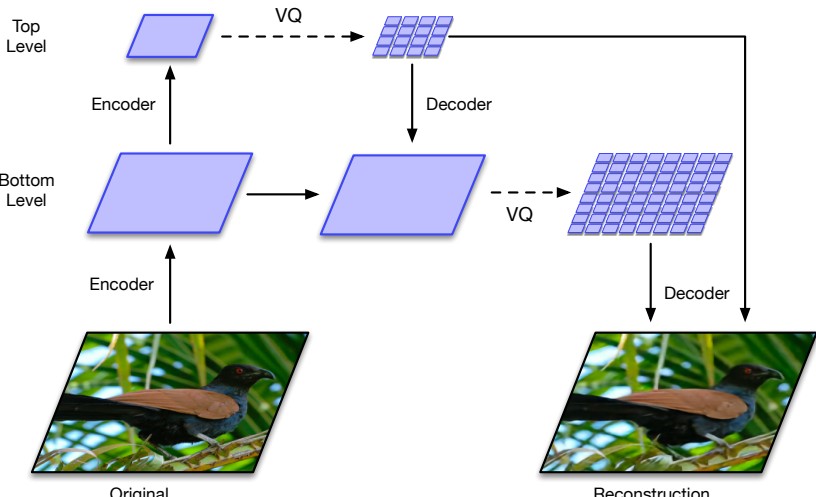

Figure 2: Overview of the architecture of our hierarchical VQ-VAE. The encoders and decoders consist of deep neural networks. The input to the model is a $256 \times 256$ image that is compressed to quantized latent maps of size $64 \times 64$ and $32 \times 32$ for the *bottom* and *top* levels, respectively. The decoder reconstructs the image from the two latent maps.

### 3.1 STAGE 1: LEARNING HIERARCHICAL LATENT CODES

As opposed to *vanilla* VQ-VAE, in this work we use a hierarchy of vector quantized codes to model large images. The main motivation behind this is to model local information, such as texture, separately from structural global information such as shape and geometry of objects. The prior model over each level can thus be tailored to capture the specific correlations that exist in that level. More specifically, the prior over the latent map responsible for structural global information, which we

---

[1] https://github.com/deepmind/sonnet/blob/master/sonnet/python/modules/nets/vqvae.py
[2] https://github.com/deepmind/sonnet/blob/master/sonnet/examples/vqvae_example.ipynb

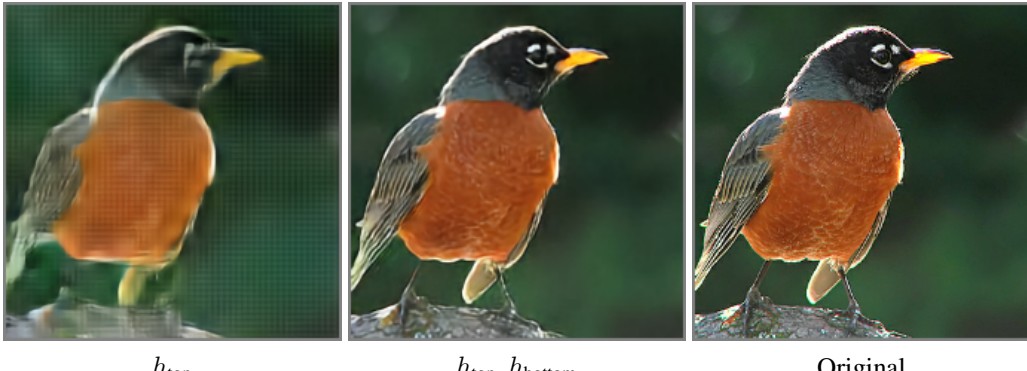

$h_{\text{top}}$      $h_{\text{top}}, h_{\text{bottom}}$      Original

Figure 3: Reconstructions from a hierarchical VQ-VAE with two latent maps (top and bottom). The rightmost image is the original. Each latent map adds extra detail to the reconstruction. These latent maps are approximately 192x and 48x smaller than the original image (respectively).

refer to as the *top* prior (see Fig. 2), can benefit from a larger receptive field of multi-headed self-attention layers to capture correlations in spatial locations that are far apart in the image. In contrast, the conditional prior model, referred to as the *bottom* prior, over latents that encode local information must have much larger resolution. As such, using as many self-attention layers as in the top-level prior is neither necessary nor practical due to memory constraints. For the prior over local information, we thus find that using a larger conditioning stack (coming from the global information code) yields more significant improvements. The hierarchical factorization also allows us to train larger models: we train each prior separately, thereby leveraging all the available compute and memory on hardware accelerators for each prior.

The structure of our multi-scale hierarchical encoder is illustrated in Fig. 2. We note that if dependencies between latent maps are such that they are strictly a compressed version of the quantized latent maps they depend on, then they would encode only redundant information that already exists in the preceding latent maps. We therefore allow each level in the hierarchy to separately depend on pixels, which encourages encoding complementary information in each latent map that can contribute to reducing the reconstruction error in the decoder.

For $256 \times 256$ images, we use a two level latent hierarchy. As depicted in Fig. 2, the encoder network first transforms and downsamples the image by a factor of 4 to a $64 \times 64$ representation which is quantized to our bottom level latent map. Another stack of residual blocks then further scales down the representations by a factor of two, yielding a top-level $32 \times 32$ latent map after quantization.

## 3.2   STAGE 2: LEARNING PRIORS OVER LATENT CODES

In order to further compress the image, and to be able to sample from the model learned during stage 1, we learn a prior over the latent codes. Fitting prior distributions using neural networks from training data has become common practice, as it can significantly improve the performance of latent variable models [2]. This procedure also reduces the gap between the *marginal posterior* and the prior. Thus, latent variables sampled from the learned prior at test time are close to what the decoder network has observed during training which results in more coherent outputs. From an information theoretic point of view, the process of fitting a prior to the learned posterior can be considered as lossless compression of the latent space by re-encoding the latent variables with a distribution that is a better approximation of their true distribution, and thus results in bit rates closer to Shannon's entropy. Therefore the lower the gap between the true entropy and the negative log-likelihood of the learned prior, the more realistic image samples one can expect from decoding the latent samples.

In the VQ-VAE framework, this auxiliary prior is modeled with a powerful, autoregressive neural network such as PixelCNN in a post-hoc second stage. More specifically, we use self-attention layers, interspersed with masked convolution blocks as proposed by [3], to model each level of the latent hierarchy as shown in Fig. 4. The top-level uses an unconditional network, and the downstream latent layers are modeled using a conditional stack that transforms the latent dependencies into spatial conditioning representations.

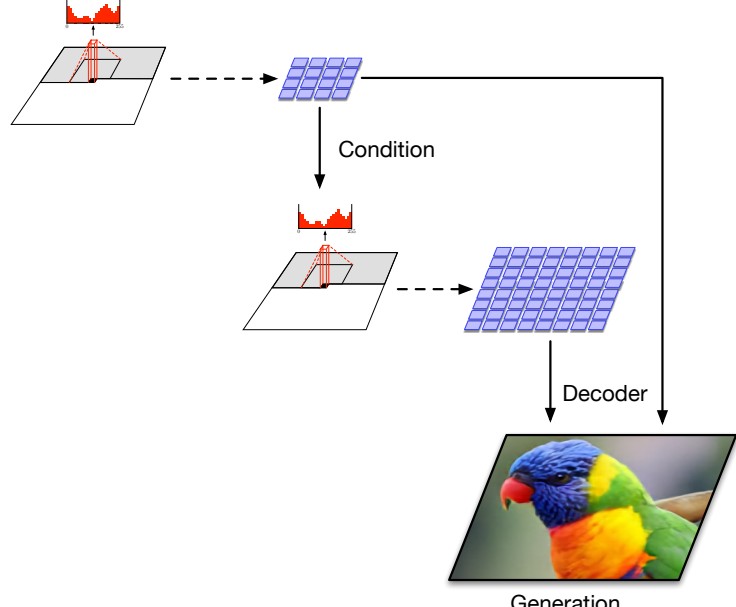

Figure 4: Multi-stage image generation: the first level of codes capture the global attributes of the image such as shape and structure, whereas the second level introduces local features such as texture and lighting refinements. Both codes are modeled with a PixelCNN, the first is conditioned on the class label, the second stage PixelCNN is conditioned on the class label and the generated codes of the first. The decoder is feed-forward, so producing the image given the two latent codes is very fast. (The example image with a parrot is generated with this model).

Our top-level prior network models $32 \times 32$ latent variables. The residual gated convolution layers of PixelCNN are interspersed with causal multi-headed attention every five layers. To regularize the model, we incorporate dropout after each residual block as well as dropout on the logits of each attention matrix. We found that adding deep residual networks consisting of $1 \times 1$ convolutions on top of the PixelSnail stack further improves likelihood without slowing down training or increasing memory footprint too much. Our bottom-level conditional prior operates on latents with $64 \times 64$ spatial dimension. This is significantly more expensive in terms of required memory and computation cost. Fortunately, as described in Sect. 3.1, the information encoded in this level of the hierarchy mostly corresponds to local features, which do not require large receptive fields as they are conditioned on the top-level prior. Therefore, we use a less powerful network with no attention layers. We also found that using a deep residual conditioning stack significantly helps at this level.

## 4 RELATED WORKS

The foundation of our work is the VQ-VAE framework of [29]. Our prior network is based on Gated PixelCNN [28] augmented with self-attention [30], as proposed in [3]. BigGAN [1] is currently state-of-the-art in FID and Inception scores, and produces high quality high-resolution images. The improvements in BigGAN were due to incorporating architectural advances such as self-attention, better stabilization methods, scaling up the model on TPUs and a mechanism to trade-off sample diversity with sample quality. In our work we also investigated how the addition of some of these elements, in particular self-attention and compute scale, improve the quality of samples of VQ-VAE models.

Recent attempts to generate high resolution images with likelihood based models include Subscale Pixel Networks of [16]. Similar to the parallel multi-scale model introduced in [19], SPN imposes a partitioning on the spatial dimensions, but unlike [19] SPN does not make the corresponding independence assumptions, whereby it trades sampling speed with density estimation performance and sample quality.

Hierarchical latent variables have been proposed in e.g. [21]. Specifically for VQ-VAE, [4] uses a hierarchy of latent codes for modeling and generating music using a WaveNet decoder. The specifics of the encoding is however different from ours: in our work the higher levels of hierarchy do not exclusively refine the information encoded in the lower levels, but they extract complementary information at each level, as discussed in Sect. 3.1. Because we are using simple, feed-forward decoders and optimizing mean squared error in the pixels, our model does not suffer from, and thus needs no mitigation for, the hierarchy collapse problems detailed in [4]. Concurrent to our work, [7] extends [4] for generating high-resolution images. The primary difference to our work is the use of autoregressive decoders in the pixel space. In contrast, for reasons detailed in Sect. 3, we use autoregressive models exclusively as priors in the compressed latent space. Additionally, the same differences with [4] outlined above also exists between our method and [7].

## 5  EXPERIMENTS

Objective evaluation and comparison of generative models, specially across model families, remains a challenge [25]. Current image generation models trade-off sample quality and diversity (or precision vs recall [22]). In this section, we present qualitative results of our model trained on ImageNet $256 \times 256$. The samples look sharp and diverse across several representative classes as can be seen in the class conditional samples provided in Fig. 6. For comparing diversity, we provide samples from our model juxtaposed with those of BigGAN-deep [1], the state of the art GAN model [3] in Fig. 5. As can be seen in these side-by-side comparisons, VQ-VAE is able to provide samples of comparable fidelity yet with much higher diversity. As mentioned previously, an important advantage of likelihood based models is that it allows assessing overfitting by comparing NLL values between training and validation sets. The NLL values reported in Table 1 for our top and bottom priors indicate that neither of these networks overfit. We note that these NLLs values are only comparable between prior models that use the same pretrained VQ-VAE encoder and decoder.

|  | Train NLL | Validation NLL |
|---|---|---|
| Top prior | 3.40 | 3.41 |
| Bottom prior | 3.45 | 3.45 |

Table 1: Train and Validation negative log-likelihood (NLL) for top and bottom prior networks. The small difference between train and validation NLL suggests that the prior networks do not overfit.

## 6  CONCLUSION

We propose a simple method for generating diverse high resolution images using VQ-VAE, combining a vector quantized neural representation learning technique inspired by ideas from lossy compression with powerful autoregressive models as priors. Our encoder and decoder architectures are kept simple and light-weight as in the original VQ-VAE, with the only difference that we propose using hierarchical multi-scale latent maps for larger images. The improvements seen in the quality of the samples are largely due to the architectural advances in the PixelCNN style priors that more accurately estimate the distribution over the latent space. In particular, using self-attention seems to be a crucial component for accurately capturing the structure and geometry of objects encoded in the top-level latent map. We also observe that the quality of our samples is correlated with the improvements in the negative log-likelihood of the model in the latent space, where small gains in likelihood often translate to dramatic improvements in sample quality. The fidelity of our best class conditional samples are competitive with the state of the art Generative Adversarial Networks, while we see dramatically broader diversity in several classes, contrasting our method against the known limitations of GANs. We believe our experiments vindicate maximum likelihood in the latent space as a simple and effective objective for learning large scale generative models that do not suffer from the shortcomings of adversarial training.

---

[3]Samples are taken from BigGAN's colab notebook in Tensorflow hub:
`https://tfhub.dev/deepmind/biggan-deep-256/1`

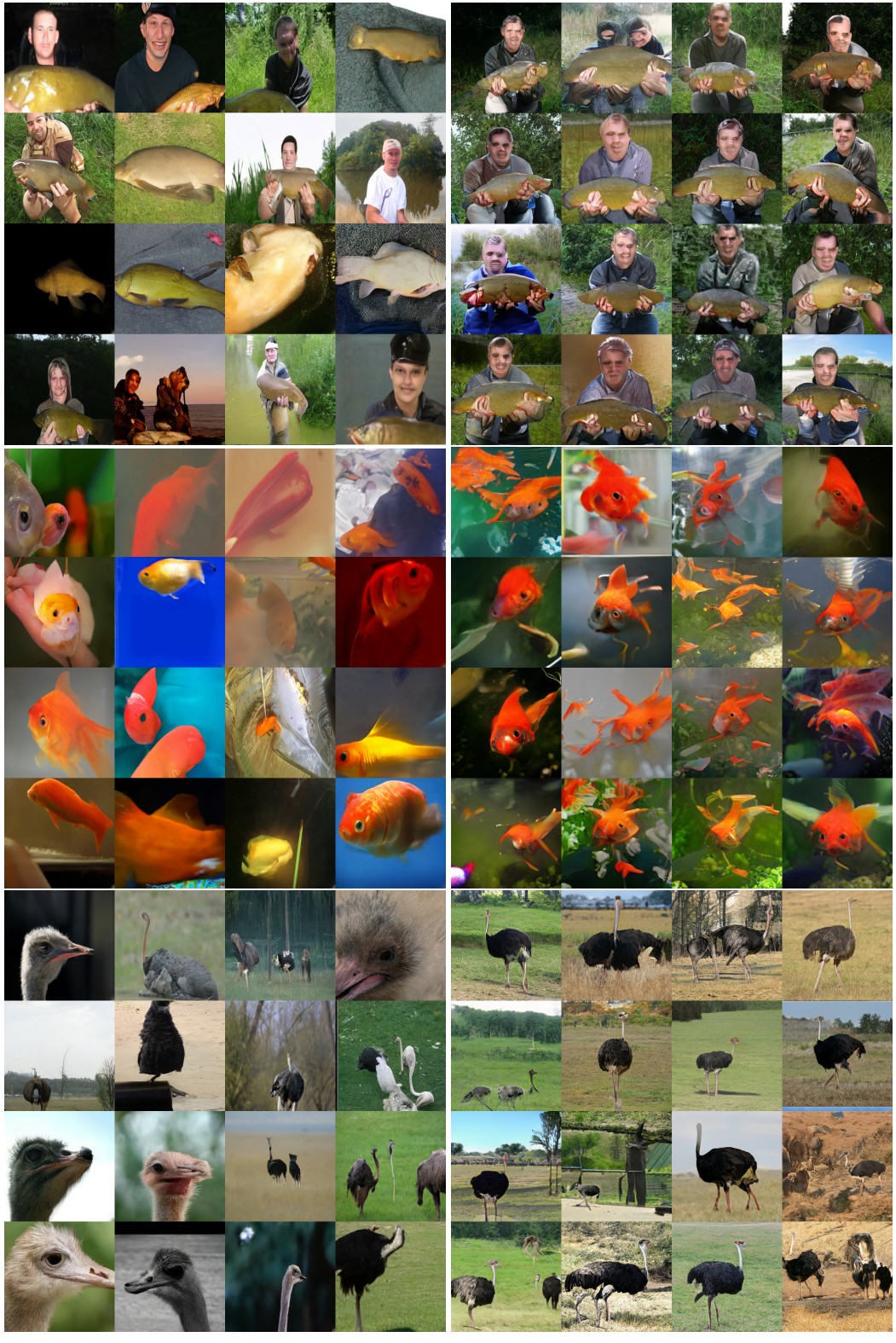

**VQ-VAE (Proposed)**                    **BigGAN deep**

Figure 5: Sample diversity comparison for the proposed method and BigGan Deep. Samples were taken for three representative classes in ImageNet: Tinca Tinca (class 0), Goldfish (class 1) and Ostrich (class 9). BigGAN samples were taken with the truncation level 1.0, to yield its maximum diversity. For Tinca, there are several kinds of samples such as top view of the fish or different kinds of poses that are absent from BigGAN's samples. Goldfish samples from VQ-VAE feature more diverse shades of orange. For Ostrich, in spite of trying many different random seeds, we were not able to get a close-up sample from BigGAN.

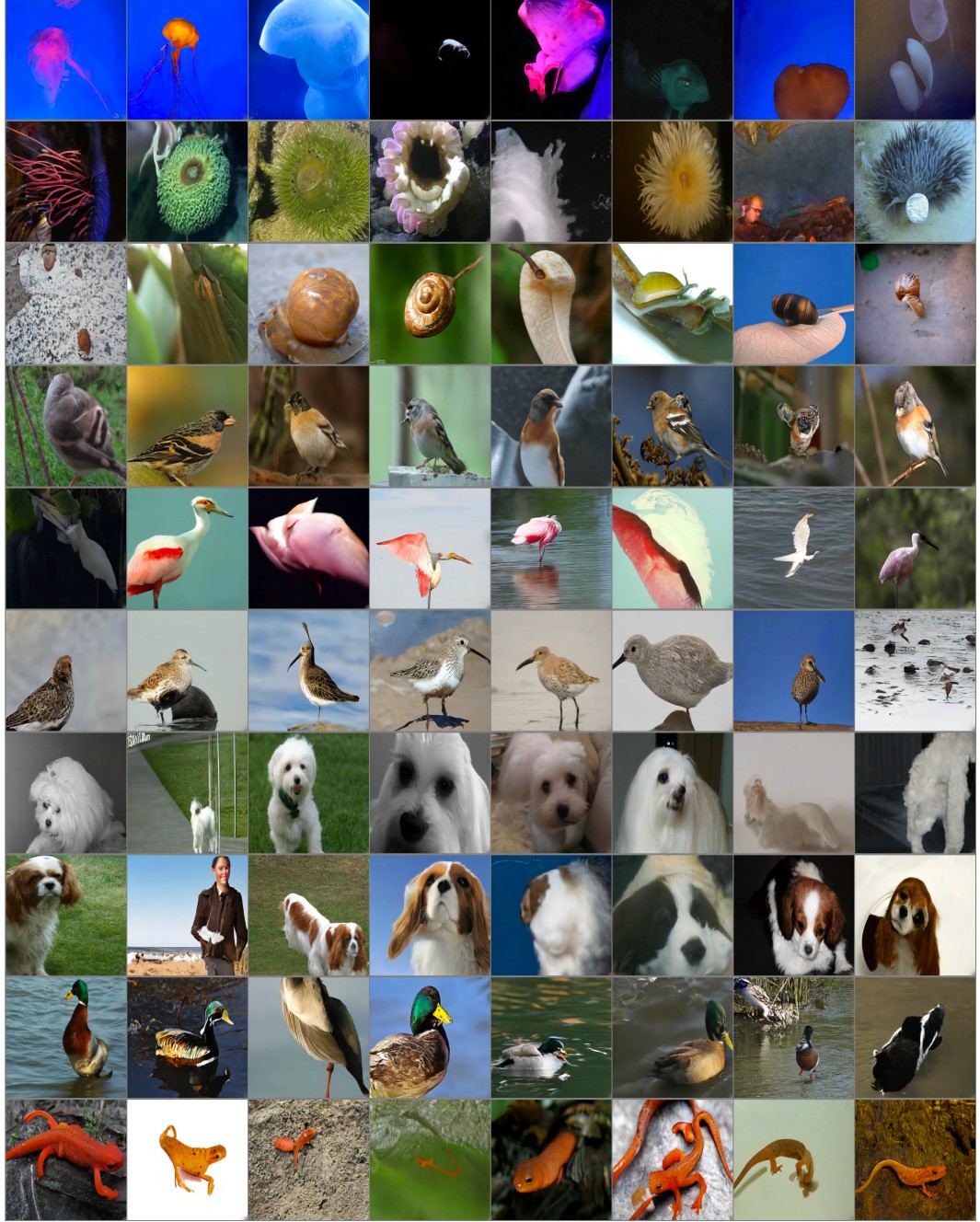

Figure 6: Class conditional random samples. Classes from the top row are: 108 sea anemone, 109 brain coral, 114 slug, 11 goldfinch, 130 flamingo, 141 redshank, 154 Pekinese, 157 papillon, 97 drake, and 28 spotted salamander.

ACKNOWLEDGMENTS

We would like to thank Danilo J. Rezende, Sander Dieleman, Jeffery Defauw, Jeff Donahue and Andy Brock for their help and feedback.

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

# A ADDITIONAL SAMPLES

We here present additional samples from our model trained on ImageNet. All these samples are taken without any cherry-picking.

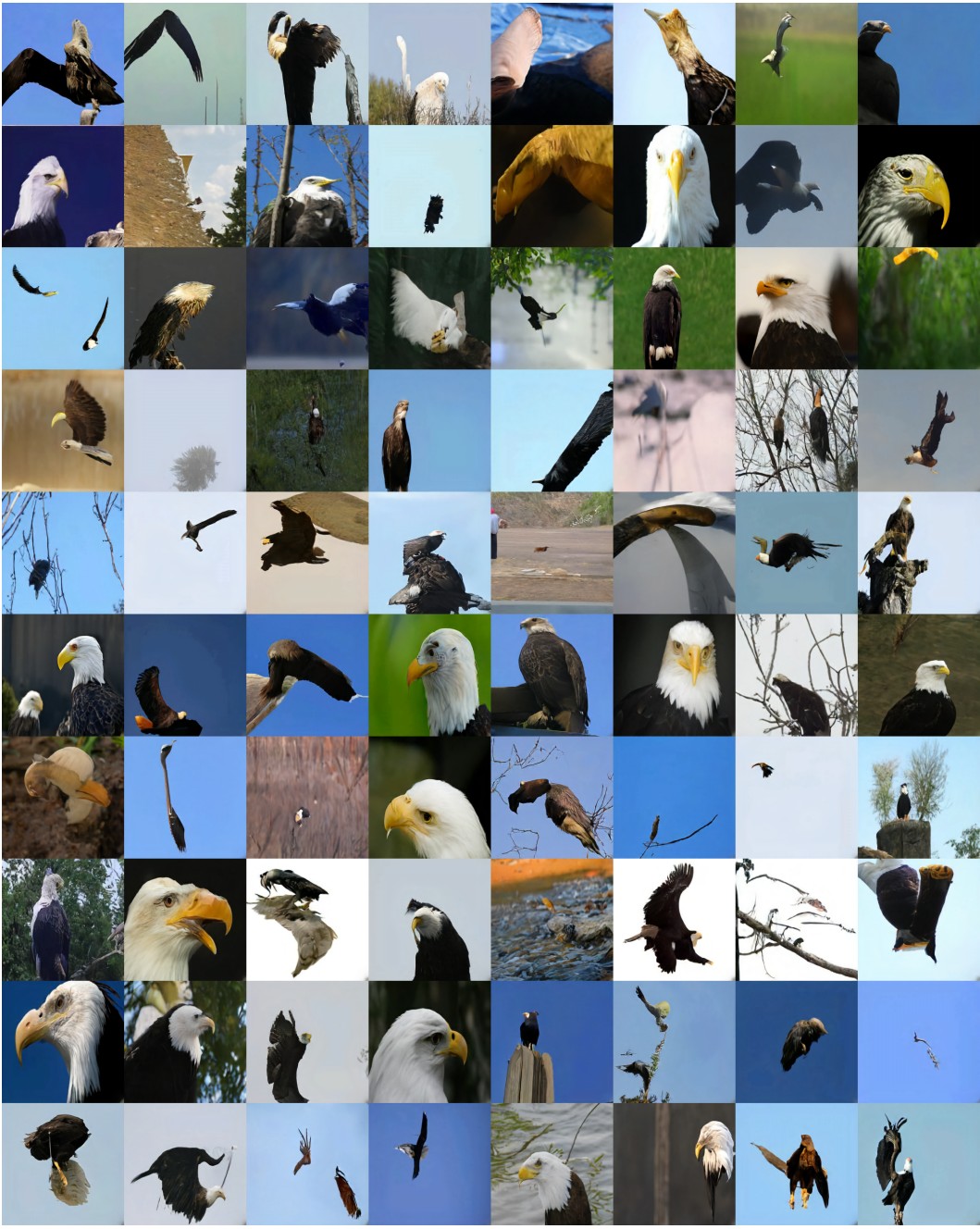

Figure 7: Samples from class 22 Bald Eagle in ImageNet

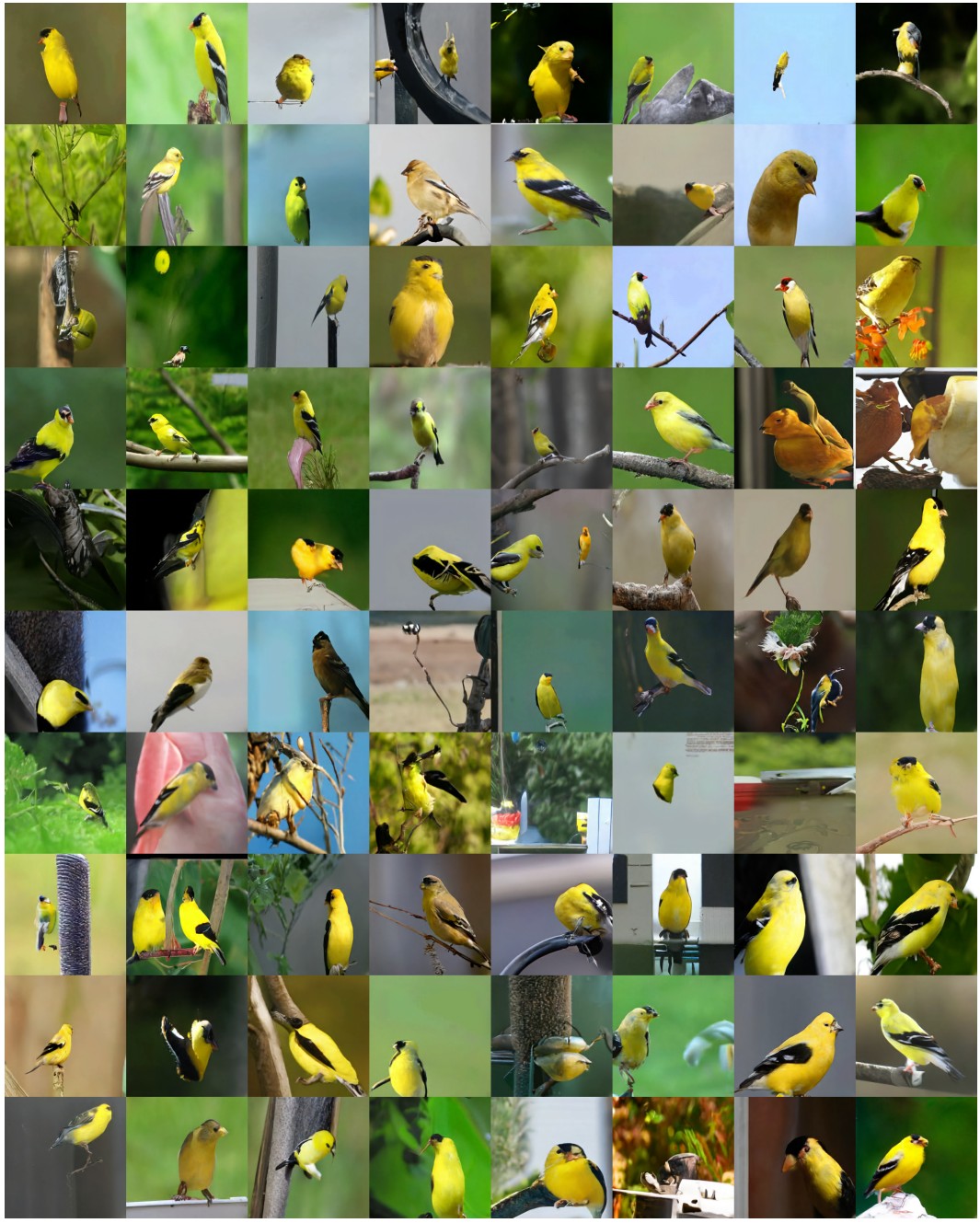

Figure 8: Samples from class 11 Gold Finch in ImageNet

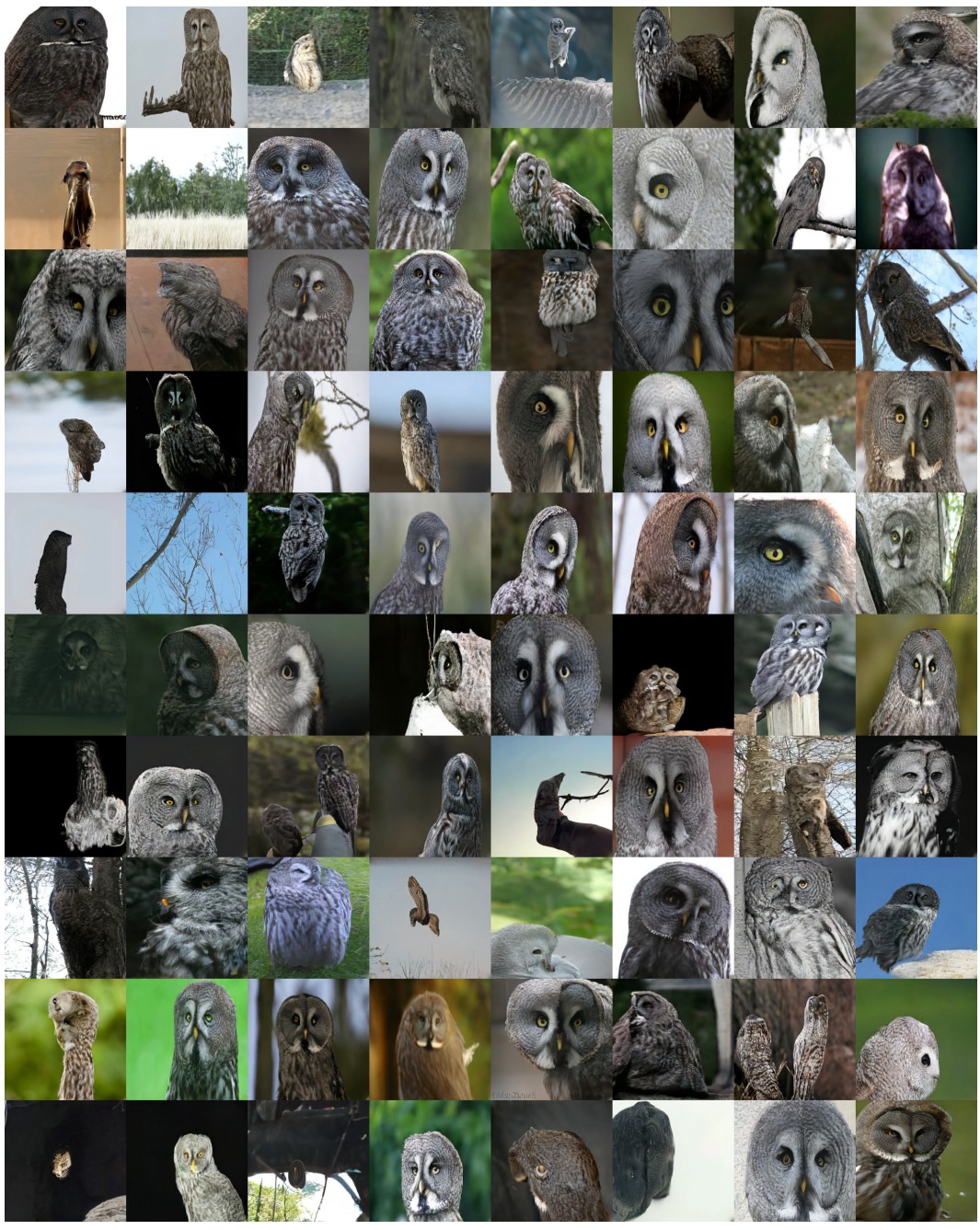

Figure 9: Samples from class 24 Grey Owl in ImageNet

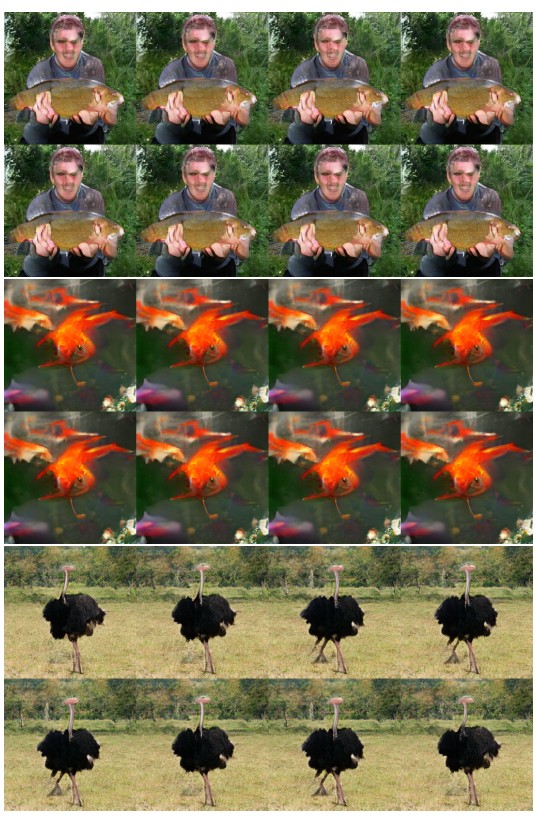

Figure 10: BigGan deep samples with truncation level 0.02 which trades diversity for sample quality.

