# OpenReview forum: "Generating Diverse High-Resolution Images with VQ-VAE"
_ICLR.cc/2019/Workshop/DeepGenStruct — DeepGenStruct 2019_

### Official Review · AnonReviewer1 · 2019-04-12
**Well written paper with great looking generated images**

**Rating:** 4
**Confidence:** 2

**Review:**

This paper proposes to use Hierarchical VQ-VAE for the purposes of large image generation. The paper is written clearly and well justified.

The authors extend the originally proposed VQ-VAE model to learn two (top & bottom) level hierarchies of images. The only con of the model is that, post-hoc PixelCNN (or PixelSnail in this paper) needs to be used to learn the prior over discrete codes in order to sample images at generation time.

Although authors claim that the model generates diverse & high quality looking it would be great to put some quantitative number on it. Doing with side-by-side samples from BigGAN and Hierarchical VQ-VAE and asking people to rate which models generated samples they prefer. As well as it would be great to see the nearest neighboring training images from the dataset according to closest distance in the embedding space.

---

### Official Review · AnonReviewer2 · 2019-04-16

**Rating:** 2
**Confidence:** 3

**Review:**

The paper proposes a method for generating diverse high resolution images with vector-quantised autoencoders (VQ-VAEs). The approach can generate images with much higher visual fidelity than the original VQ-VAE paper via two main ingredients: (1) hierarchical multi-scale latent maps and (2) PixelSNAIL instead of PixelCNN.

The motivation and the contributions of this paper are very similar to De Fauw et al., 2019 (Hierarchical Autoregressive Image Models with Auxiliary Decoders; https://arxiv.org/abs/1903.04933), in that De Fauw et al. also used hierarchical VQ-VAEs (specifically, 2-layer) to generate high fidelity images. Also their autoregressive priors are closely related to PixelSNAIL. I'm assuming the authors were not aware of this paper, as they did not cite it. De Fauw et al. report IS/FID/Test NLL, none of which this paper reports. Given De Fauw et al. 2019 was submitted to arxiv on March 9th, this should be considered concurrent work, but should be cited in the revision.

Pros
- Clear exposition and motivation
- High fidelity and diversity in the generated images

Cons
- No test NLL comparisons with other likelihood based approaches (e.g. SPN, Parallel Multiscale)
- I'd have liked to see Inception/FID results from the proposed model (as done by De Fauw et al., 2019).

---

### Decision · Program_Chairs · 2019-04-19
**Acceptance Decision**

**Decision:**

Accept

**Comment:**

The generated images are impressive but as the reviewers note, it would be good to have quantitative comparison against existing methosd.